20

# A Global High-Resolution Hydrological Model to Simulate the Dynamics of Surface Liquid Reservoirs: Application on Mars

Alexandre Gauvain<sup>1</sup>, François Forget<sup>1</sup>, Martin Turbet<sup>1,2</sup>, Jean-Baptiste Clément<sup>1</sup>, Lucas Lange<sup>1</sup>, and Romain Vandemeulebrouck<sup>1</sup>

Correspondence: Alexandre Gauvain (alexandre.gauvain@lmd.ipsl.fr, alexandre.gauvain.ag@gmail.com)

**Abstract.** Surface runoff shapes planetary landscapes, but global hydrological models often lack the resolution and flexibility to simulate dynamic surface water bodies beyond Earth. Recent studies of Mars have revealed abundant geological and mineralogical evidence for past surface water, including valley networks, crater lakes, deltas and possible ocean margins dating from late Noachian to early Hesperian times. These features suggest that early Mars experienced periods allowing liquid water stability, runoff and sediment transport. To investigate where surface water could accumulate and how it may have been redistributed, we developed a global high-resolution (km-scale) surface hydrological model. The model uses a pre-computed hydrological database that maps topographic depressions, their spillover points, hierarchical connections between basins, and lake volume-area-elevation relationships. This database approach greatly accelerates simulations by avoiding repeated geomorphic processing. The model dynamically forms, grows, merges and dries lakes and putative seas without prescribing fixed coastlines, by transferring water volumes between depressions according to their storage capacities and overflow rules. We explore model behavior over the present-day Mars' topography measured by MOLA (Mars Orbiter Laser Altimeter) topography for a range of evaporation rates and total water inventories expressed as Global Equivalent Layer (GEL). Simulations are iterated to steady state under the assumption that precipitation balances evaporation plus overflow. The model outputs the extent and depth of surface water bodies and identifies main drainage pathways using overflow fluxes as runoff indicators. Results show a transition toward a contiguous northern ocean between low (1-10 m) GEL values and increasing concentration of water in northern lowlands and major impact basins at higher GEL. We discuss the model's limitations, including its dependence on topography and the absence of subsurface flows, and propose future improvements. This framework provides a quantitative tool to link preserved geomorphology with plausible past hydrological states. Future work will couple the model with a 3D global climate model into a Planetary Evolution Model (PEM) to study transient water redistribution and climate-hydrology feedbacks.

<sup>&</sup>lt;sup>1</sup>Laboratoire de Météorologie Dynamique, CNRS, Sorbonne Université, Paris, France

<sup>&</sup>lt;sup>2</sup>Laboratoire d'Astrophysique de Bordeaux, Université de Bordeaux, Bordeaux, France

#### 1 Introduction

Surface runoff is a key process in the interaction between climate, hydrology and geomorphology, shaping planetary landscapes (Salles et al., 2023, 2020). On Earth, several global hydrological models have been developed to study the water cycle (e.g., Sood et al., 2015) and its interactions with the climate. However, these global models typically operate at low resolution, and representing hydrological processes at higher resolution remains a significant challenge (van Jaarsveld et al., 2025). Additionally, these models are typically solved only over the continental domain, with oceans prescribed as fixed boundary conditions rather than being dynamically simulated. While Earth-based models are capable of simulating the dynamic evolution of continental water bodies in response to climatic changes, their direct application to other planets is limited, since the presence, extent, and variability of oceans or large lakes over paleoclimatic timescales remain highly uncertain. Moreover, geomorphological evidence of fluvial activity is often preserved at a level of detail far higher than the resolution of these models, further limiting their deployment for planetary studies.

On Mars, high-resolution observations reveal extensive evidence for past surface water flow (Carr and Clow, 1981; Seybold et al., 2018). Valley networks are much more numerous than previously thought, with higher drainage densities and strong evidence of sustained precipitation and surface runoff (Hynek et al., 2010). Detailed geomorphological analyses suggest that many valleys were carved by surface runoff producing V-shaped profiles, and subsequently modified by sapping processes that widened their cross-sections (Williams and Phillips, 2001; Irwin et al., 2005a). Estimated formative discharges for some Martian valleys are comparable to — and in some cases exceed — those of terrestrial precipitation-fed networks, consistent with episodic flood-like flows (Kite et al., 2019; Irwin et al., 2005a). Additional, evidences point to larger bodies of water in Mars' past. Paleoshorelines and the distribution of deltas in the northern lowlands have been interpreted as indicating ancient oceans and large lakes (Di Achille and Hynek, 2010; Zuber, 2018; Li et al., 2025; Perron et al., 2007; Ivanov et al., 2017; Sholes et al., 2021; Citron et al., 2018; Head et al., 1999). The identification of open- and closed-basin lakes (Goudge et al., 2016; Fassett and Head, 2008) highlights the role of topographic depressions/craters in capturing and storing water. These lakes often exhibit inlet and outlet valleys, providing evidence of hydrological connectivity. Within craters and along the dichotomy boundary (transition between the old, high, cratered southern highlands and the younger, low northern plains), numerous preserved deltas attest to sustained sediment transport and deposition into standing water bodies (Fassett and Head, 2005; Palucis et al., 2014; Di Achille and Hynek, 2010). The widespread occurrence of hydrated minerals — including phyllosilicates, hydrated sulfates and hydrated silica — provides further evidence that liquid water was present on early Mars (Bibring et al., 2006; Carter et al., 2015; Ehlmann et al., 2011; Beck et al., 2025).

Despite this quantity of geomorphological and mineralogical evidences, the origin and maintenance of Martian fluvial activity remain debated and multiple processes may have contributed to valley network formation (Bouley et al., 2009). Proposed mechanisms include precipitation-fed surface runoff during transient warm/wet intervals (Craddock and Howard, 2002; Irwin et al., 2005b; Mangold et al., 2004), prolonged or episodic water flow (Carr and Malin, 2000), groundwater sapping possibly sustained by geothermal or magmatic activity (Carr and Chuang, 1997; Williams and Phillips, 2001; Carr and Malin, 2000);

65

Malin and Carr, 1999; Morgan and Head, 2009; Wordsworth et al., 2013), and subglacial or snowmelt-driven processes in colder climates (Grau Galofre et al., 2020).

To resolve these questions, it is critical to identify where water could accumulate and act as active reservoirs within Mars' hydrological cycle. Mapping the locations and sizes of possible surface water bodies, and quantifying exchanges between surface and atmosphere, are necessary steps to use geomorphological observations to constrain climatic conditions (Matsubara et al., 2011; Kamada et al., 2020, 2021; Boatwright and Head, 2019). To understand the climate responsible for the formation of water-related geomorphological features – such as valley networks, lakes, oceans and sedimentary deposits – a global high-resolution hydrological model is needed to simulate the surface water distribution and compare it with the observed geomorphological features. This new hydrological model will provide a realistic distribution of surface water bodies, which can be used as input for 3D Global Climate Models (GCMs) to study the interactions between climate and hydrology (e.g., Turbet and Forget, 2021; Wordsworth et al., 2015).

This paper presents a new global high-resolution hydrological model designed to simulate the spatial distribution of surface water reservoirs and their connectivity for a given global topography map. The first section describes the model assumptions and framework, particularly the use of a hierarchical depression graph, the construction of the hydrological database and pre-computed hydrological functions. The implementation of the model is then explained, covering the algorithms for water overflow, evaporation, and the iterative scheme for reaching steady state. Then we present an application on Mars by using his current high-resolution topographic map. This section details the topographic data used to construct the hydrological database and the focused area chosen to present the results. The results' section analyse the model outputs, such as water depth, volume distribution and flow pathways. Finally, the discussion highlights the current model limitations, including its dependence on topography and the absence of subsurface flows, and proposes future improvements.

#### 2 Model implementation

## 75 2.1 General model description

This hydrological model is designed to simulate the routing and accumulation of surface water over a given topography. Its core principle is that water flows across the landscape until it encounters depressions where it can be stored, allowing for the formation of lakes or oceans. A key feature of the model is that these lakes/oceans are not fixed boundaries: they can completely evaporate under arid conditions and reappear during wetter periods. When a lake reaches its maximum storage capacity, the model transfers the overflow to downstream depressions. This mechanism enables the transfer of water volumes between reservoirs provides a way to trace drainage pathways and reconstruct watercourses. To simulate these dynamics efficiently, the model makes several simplifying assumptions. In principle, surface-water flow is governed by the conservation of mass and momentum equations integrated over flow depth. However, solving these equations at high resolution and over large domains is computationally prohibitive. Instead, our approach assumes that water follows the steepest topographic slope until it reaches an unfilled depression.

The hydrological model is based on the existing frameworks of Barnes et al. (2020, 2021) and Callaghan and Wickert (2019). This framework is specifically designed to handle complex terrains with numerous depressions, like craters for instance (Figure 1a). It uses a binary tree data structure, known as the depression hierarchy, where each node represents either a watershed of a depression ("leaf" depression) or a meta-depression formed by the merging of two filled depressions (Figure 1b). The depressions are interconnected in a network that defines the water flow paths within the binary tree (Figure 1c). This structure significantly enhances computational efficiency, as the connections between depressions are only determined once during the pre-processing stage. Additionally, pre-computed functions link the lake elevation, water volume, and lake area for each depression, enabling quick estimation of lake surface or elevation from water volume. All of this information are gathered in a hydrological database that will be described in Section 2.2. The implementation of the hydrological model is described in Section 2.3.

**Figure 1.** Representation of the water cycle on a conceptual topography with 4 leaf depressions. (a) 3D block represents the water flow from upstream to downstream. The links between the depressions, which represent streams, are symbolized by full blue curves. The sky blue dotted line represents the lake elevation where a depression No.1 can merge with a depression No.2 and create a new depression No.5. The dark blue dotted line represents the lake elevation where the depression No.5 merge with the depression No.3. The merge lines (blue and black dotted lines) are projected on the cross-section of the 3D block diagram (b). (c) Binary tree of the depression hierarchy. Nodes No.1-4 are leaf depressions and nodes No.5-7 are meta-depressions. The black lines represent the merge into a new depression. The dotted arrows show the link with downstream depression. Adapted from Barnes et al. (2020).

A word of cautious: the model is originally designed for watershed-scale simulations, and a key limitation for its planet-scale applicability is that we assume that water transfer between depressions occurs instantaneously. This simplification may lead

100

to a slight underestimation of evaporation during transfer and neglects potential transient storage effects. Furthermore, each node in the binary tree represents the watershed of a depression with its own storage capacity, meaning that intra-watershed water routing is not explicitly simulated. Infiltration and subsurface flow processes are also not included at this stage of model development. Some of these limitations can be mitigated through post-processing, as described in Section 2.4.

#### 2.2 Pre-computed hydrological database

The hydrological database aims to store all the information derived from the high resolution map that needs to be computed only once. This operation maintains an acceptable computation speed since the hydrological model will only have to load the database and directly use this data. To build this hydrological database, we used the Depression Hierarchy algorithm (Barnes et al., 2020) to calculate the binary tree (Figure 1c, Section 2.2.1) and we pre-computed the depression information (Section 2.2.2) and functions that establish the link between volume lake, elevation lake and surface area lake for each depression (Section 2.2.3). All the information saved in the hydrologic database are listed in the Table 1.

## 2.2.1 Hierarchical depression graph

The depression hierarchy algorithm is a powerful method for analyzing and filling depressions from a Digital Elevation Model (DEM) to create a hierarchical depression graph across complex landscapes. The algorithm, proposed by Barnes et al. (2020), is divided into four stages: (1) ocean identification, (2) pit cell identification, (3) depression assignment, and (4) hierarchy construction. As mentioned previously, the model must allow waterbodies to evolve dynamically. The ocean identification stage, designed for Earth, is then not necessary since this step fixes the elevation of the oceans. To bypass this feature of the algorithm, the maximum ocean elevation was fixed at the highest cell of the DEM. This setting allows that a large amount of water on the planet can create a planetary scale ocean. The process begins by computing the water flow direction using the simple D8 algorithm (O'Callaghan and Mark, 1984; Barnes, 2016). For each cell of the MOLA dataset, the water flow direction is defined by following the steepest slope calculated from the 8 neighboring cells. This flow direction map allows to identify the depressions, i.e. DEM cells without outlet, i.e. all 8 neighboring cells which have an altitude equal to or greater than their own. After locating these depressions, the algorithm assigns each DEM cell with a pit cell to delimit the watershed. The Depression Hierarchy algorithm gives a label map of the depression identifiers (ID) representing the watershed area for each leaf depression. This map is saved in the hydrological database. At this step, the algorithm fills each depression until the spillover point, which is the lowest elevation point where water can overflow into neighboring terrain (Barnes et al., 2014). The spillover point of the watershed is identified as being the lowest elevation point on the watershed boundary. Then, a hierarchical structure is built on how the depressions are nested or connected to another one. Smaller depressions that fill first may overflow into adjacent depressions, establishing a parent-child relationship. The algorithm then links each depression to its parent depression through its spillover point, creating a binary tree where depressions are ranked according to their order of filling. Each depression i has a parent depression  $P_i$ , a sibling depression  $S_i$  and a targeted downstream depression  $D_i$ . The meta-depression i (merge of two depressions) has a right child  $R_i$  and a left child  $L_i$ . For instance, the meta-depression No.5 (Figure 1b-c.) has the depression No.6 as parent, depression No.3 as the sibling and targeted downstream depressions, and the

**Table 1.** Maps and parameters saved in the hydrological database (NetCDF file). Parameters are divided into four parts: (1) Maps, (2) Depression Hierarchy Graph (Section 2.2.1), (3) Depression Information (Section 2.2.2) and (4) Hydrological Functions (Section 2.2.3). Each depression is referenced by a unique identifier (ID).

|                                  | Parameter            | Description                                  | Unit   |
|----------------------------------|----------------------|----------------------------------------------|--------|
| Maps                             | DEM                  | Digital Elevation Model                      | m      |
|                                  | Watershed            | Label map of the leaf depression watershed   | ID     |
| Depression<br>Hierarchy<br>Graph | i                    | Depression                                   | ID     |
|                                  | $P_i$                | Parent of the depression $i$                 | ID     |
|                                  | $S_i$                | Sibling depression                           | ID     |
|                                  | $D_i$                | Downstream depression                        | ID     |
|                                  | $R_i$                | Right child depression                       | ID     |
|                                  | $L_i$                | Left child depression                        | ID     |
|                                  | $Z_i^{max}$          | Elevation of the spillover point             | m      |
|                                  | $Z_i^{min}$          | Elevation of the merge point of the children | m      |
|                                  | $A_i^w$              | Watershed area                               | $m^2$  |
|                                  | $V_i^{max}$          | Maximum water volume                         | $m^3$  |
| Depression                       | $\lambda_i^{sp}$     | Longitude of the spillover point             | degree |
| Information                      | $\varphi_i^{sp}$     | Latitude of the spillover point              | degree |
|                                  | $\lambda_i^{min}$    | Minimum longitude of watershed extent        | degree |
|                                  | $\lambda_i^{max}$    | Maximum longitude of watershed extent        | degree |
|                                  | $arphi_i^{min}$      | Minimum latitude of watershed extent         | degree |
|                                  | $\varphi_i^{max}$    | Maximum latitude of watershed extent         | degree |
| Hydrological                     | $V_i^l = f_i(Z_i^l)$ | Lake volume as a function of lake elevation  | $m^3$  |
| Functions                        | $A_i^l = f_i(Z_i^l)$ | Lake area as a function of lake elevation    | $m^2$  |

depressions No.1 and No.2 as right and left children. The last node at the top of the binary tree represents the whole planet. This algorithm is designed to generate a hierarchical depression graph at watershed scale. For a planetary (spherical) context, some modifications are required to ensure flow continuity across the globe. Specifically, the algorithm is adjusted to account for the east-west continuity, linking the two opposite sides of the DEM. This modification allows depressions at the interface of the DEM boundaries to be identified and connected.

# 2.2.2 Depression information

135

Once the hierarchical depression graph is established, several parameters in the hydrologic database (Table 1) are computed to characterize the depressions and the associated watershed. The maximum and minimum water elevations in the depression,  $Z_i^{max}$  and  $Z_i^{min}$ , are directly given by the depression hierarchy algorithm.  $Z_i^{max}$  corresponds to the elevation of the spillover

40 point (Figure 1a). For the leaf depression, Z<sub>i</sub><sup>min</sup> is the minimal elevation given by topography elevation in the watershed while, for the meta-depression, it is the elevation of the spillover point of his two children (Z<sub>Ri</sub><sup>max</sup> and Z<sub>Li</sub><sup>max</sup>). Using the label map, the watershed area A<sub>i</sub><sup>w</sup>, which represents the surface contributing to a given depression, can be computed by summing the depression cell areas. The maximum volume of water that each depression can contain, V<sub>i</sub><sup>max</sup>, is computed from the elevation topography map, the watershed area and the water depth below Z<sub>i</sub><sup>max</sup>. For the meta-depressions, V<sub>i</sub><sup>max</sup> is computed with the water depth between Z<sub>i</sub><sup>min</sup> and Z<sub>i</sub><sup>max</sup>. λ<sub>i</sub><sup>sp</sup> and φ<sub>i</sub><sup>sp</sup> specify the coordinates (longitude and latitude) of the spillover point. λ<sub>i</sub><sup>min</sup>, λ<sub>i</sub><sup>max</sup>, φ<sub>i</sub><sup>min</sup> and φ<sub>i</sub><sup>max</sup> are the grid coordinates delimiting the extent of the watershed depression. These coordinates are mainly used to focus on a specific depression, decrease the computation time of constructing hydrological functions and facilitate the post-processing.

# 2.2.3 Pre-computed hydrological lake functions

To efficiently convert the volume stored by the depression into lake water levels or lake surface areas, we pre-computed these relationships as functions of the lake elevation. These functions characterize the evolution of three lake-related metrics: the lake volume  $V_i^l$ , the lake surface area  $A_i^l$  and the lake elevation  $Z_i^l$ . For each depression and each metric, a value is computed at every 10% of the water elevation, between  $Z_i^{min}$  and  $Z_i^{max}$ . The hydrological functions are stored as look-up tables to be linearly interpolated in the hydrological database (Table 1) and loaded by the hydrological model when the simulation starts.

A linear interpolation is used to estimate the lake area  $A_i$  and the lake elevation  $Z_i$  based on a given lake volume  $V_i$ . The algorithm searches for the appropriate interval  $[V_{i,k}^l, V_{i,k+1}^l]$  in the pre-computed hydrological function such that the index k is the largest index satisfying  $V_{i,k}^l 

180

185

depressions and their hierarchical relationships. Next, the model connects each leaf depression with the corresponding grid cell in the GCM. The first step of the hydrological model is an initialization stage. A Global Equivalent Layer (GEL) is specified as a fixed-depth layer of water allover the planet. The initialization can also be performed by injecting a specified amount of water at a georeferenced point (longitude and latitude). In order to distribute this amount of water, a volume is injected in each leaf depression following the equation:

$$V_i = A_i^w \times GEL \tag{3}$$

Then, all leaf depressions contain a volume of water and are therefore considered active  $(a_i = 1)$ . The algorithm checks if the water volume  $V_i$  of the leaf depression is higher than the maximum volume  $V_i^{max}$  and will handle the excess volumes  $V^{excess}$ . Once the excess volumes are redistributed following the recursive subroutine described in Section 2.3.2, the lake areas are computed by interpolation using the hydrological lake functions (Section 2.2.3). Then, the location and area of lakes can be used as inputs to the GCM, indicating the percentage of water coverage for each GCM grid cell. The GCM then provides the precipitation and evaporation rates. The retrieved precipitation and evaporation data are then used to update the water distribution in the hydrological model. This iterative process continues, with the model recalculating the water surface area and feeding it back into the GCM, ensuring a dynamic and interactive simulation of the hydrological and climatic systems.

To test and validate the hydrological model, conceptual cases are investigated by replacing the GCM simulation by a simple process with a constant evaporation rate E (m·s<sup>-1</sup>) and homogeneous precipitation of the total evaporated volume. This process allows to conserve the mass of water in the system and to reach an equilibrium of the water surface reservoirs. After the initialization step (Equation 3), the hydrological model uses an iterative scheme to reach a steady state. One iteration of the model corresponds to the following scheme. First, the potential evaporated volume  $V_i^{Ep}$  (m<sup>3</sup>) is computed for each depression i based on its active state  $a_i$ , the evaporation rate E and water surface area  $A_i$  (m<sup>2</sup>):

190 
$$V_i^{Ep} = \begin{cases} A_i E \Delta t & \text{if } a_i = 1\\ 0 & \text{if } a_i = 0 \end{cases}$$
 (4)

where  $\Delta t$  is the adaptive time step (s) of the simulation. In function of the value of  $V_i^{Ep}$  and the available volume  $V_i$ , the model computes the real evaporated volume  $V_i^{Er}$  that each depression i is able to evaporate (Section 2.3.3). In the case where the available volume is lower than  $V_i^{Ep}$ , the adaptive time step  $\Delta t$  is decreased to ensure that the relative difference between  $V_i^{Ep}$  and  $V_i^{Er}$  is negligible (less than 1%). To ensure mass conservation, the precipitation rate P (m.s<sup>-1</sup>) is then calculated using the total of real evaporated volume  $V_i^{Er}$  and the watershed areas  $A_i^w$  of active depressions:

$$P = \frac{\sum_{i=1}^{N} a_i V_i^{Er}}{\Delta t \sum_{i=1}^{N} a_i A_i^{w}}$$
 (5)

Next, the water volume change  $\Delta V_i$  is computed for each active depression i by combining the water balance equation (Howard, 2007; Matsubara et al., 2011, 2013) with the adaptive time step  $\Delta t$ :

$$\Delta V_i = \Delta t \left[ (A_i^w - A_i) P \beta + A_i P - A_i E \right] \tag{6}$$

where  $\beta$  is the fraction of precipitation that can runoff. In this conceptual study, it is assumed that there is no infiltration ( $\beta = 1$ ). If  $\Delta V_i < 0$ , the absolute value of  $\Delta V_i$  is evaporated from the depression following the process describes in the Section 2.3.3. Conversely, if  $\Delta V_i > 0$ , the depression receives the water volume  $\Delta V_i$ . If  $V_i + \Delta V_i$  is higher than  $V_i^{max}$ , the excess volume is redistributed following the water overflow algorithm (Section 2.3.2). Once the water volume change is processed and the water volumes  $V_i$  are updated for all depressions, the model recalculates the lake elevation  $Z_i$  and lake surface area  $A_i$  using the hydrological lake functions (Section 2.2.3). This iterative process continues until the system reaches a steady state when the water volume  $V_i$  and lake surface area  $A_i$  in each depression remains constant. More precisely, equilibrium is assumed when the relative change in water volume and surface area in each active depression between two successive iterations is less than 0.1%, and no new overflow events are triggered anywhere in the planet.

## 2.3.2 Overflow algorithm

230

- The hierarchical relationship between nodes (depression, sibling, and parent nodes) governs how excess water is redistributed. The main challenge in this model is managing excess water that accumulates when a reservoir exceeds its storage capacity. The process follows a set of rules and priorities to distribute effectively the excess water,  $V^{excess}$ , based on a recursive subroutine following three successive steps (Barnes et al., 2021):
- (1) If the water volume in the considered depression  $V_i$  is less than its maximum capacity  $V_i^{max}$ , the remaining available volume  $V_i^{avail} = V_i^{max} V_i$  is used to store a part or the entire excess volume. In case where  $V^{excess}$  is smaller than  $V_i^{avail}$ , the reservoir stores all the excess water, and  $V^{excess}$  becomes equal to 0. Conversely, when  $V^{excess}$  is larger than  $V_i^{avail}$ , the remaining excess water is calculated as:  $V^{excess} = V^{excess} V_i^{avail}$  and the depression fills up ( $V_i^{avail} = 0$  and  $V_i = V_i^{max}$ ). In the latter case, this excess water will be redirected to another storage location according to the steps (2) and (3).
- 220 (2) If the considered depression is full  $(V_i = V_i^{max})$ , the next step is to check if the redirection of the excess water to the sibling depression  $S_i$  is possible. As a reminder, the sibling depression is a neighboring depression at the same level of the hierarchy. This redirection of the excess water is only possible if the sibling has available space  $V_{S_i}^{avail} > 0$  and has already water in the depression  $V_{S_i} > 0$ . If the water volume of the sibling depression  $V_{S_i}$  is equal to 0, then this means that the excess volume must be transferred to an active downstream depression (i.e. a depression at a lower level of the hierarchy). The excess volume is transferred to the downstream depression  $D_i$ . Depending on these conditions, the recursive process restart at the step (1) in the sibling depression  $S_i$  or downstream depression  $D_i$  with the excess volume computed in step (1).
  - (3) If the volume depression  $V_i$  of the considered depression i and its sibling  $V_{S_i}$  are full, the last option is to transfer the excess water to the parent depression  $V_{P_i}$ , which is higher up in the hierarchy. The active state of the considered depression  $a_i$  and the sibling depression  $a_{S_i}$  is set to 0. The active state of the parent depression  $a_{P_i}$  switches to 1. The recursive process restarts at the step (1) in the parent depression  $P_i$  with the excess volume computed in step (1).

235

240

The excess volume of water going out from each depression is accumulated in  $Q_i^{out}$ . This recursive process continues, allowing the model to simulate water moving up within the depression hierarchy graph. The recursive loop finish when  $V^{excess}$  is equal to 0.

To optimize the computational efficiency of the hydrological model, we implemented a bypass mechanism. For a given depression, the final depression where  $V^{excess}$  reaches zero can be stored in memory. Subsequently, if another  $V^{excess}$  is introduced into the same initial depression, the algorithm bypasses intermediate steps and directly routes the new  $V^{excess}$  to the previously identified final depression. This approach significantly reduces redundant calculations and accelerates the redistribution of water across the depression hierarchy, particularly in scenarios of repetitive water flow patterns. This option can be turned on or off in the hydrological model. The activation of this bypass mechanism have a negative impact since it does not make the overflow  $Q_i^{out}$  possible to compute. However, once the simulation has converged to a steady state with the bypass enabled, this option can be disabled for one additional iteration to accurately recalculate the overflow fluxes for each depression.

## 2.3.3 Evaporation algorithm

To simulate the evaporation process, the hydrological model uses a hierarchical evaporation scheme. Considering a depression i and a potential evaporated volume  $V_i^{Ep}$  (Section 2.3.1, Equation 4) in this same depression, the algorithm traverses the depression hierarchy graph by level, i.e. the depressions located below depression i (i.e., right child  $R_i$  and left child  $L_i$ , Figure 1b-c). When a sufficient amount of water is present in depression i and its underlying depressions, the real evaporated water volume  $V_i^{Er}$  equals the potential water volume to evaporate. Otherwise, when the amount of water is insufficient, the real evaporated water volume  $V_i^{Ep}$  becomes less than the potential evaporated volume  $V_i^{Ep}$ . The algorithm continues to traverse the hierarchy by level until it reaches a leaf depression with no water or it evaporates all the potential evaporated volume  $V_i^{Ep}$ . The evaporation process can be expressed as:

$$V_i^{Er} = \min\left(\sum_k V_k^E, V_i^{Ep}\right) \tag{7}$$

where  $\sum_k V_k^E$  is the sum of the evaporated volume in all depressions k that are below the depression i, depression i included.

# 255 2.4 Model outputs and post-processing

The simulation outputs are saved in a NetCDF file and can be saved at each iteration, depending on the frequency specified by the user, or only at the end of the simulation. The saved data include the lake surface elevation  $Z_i$ , the area covered by the lake  $A_i$ , the volume of water stored in the depression  $V_i$ , the active state  $a_i$  (indicating whether it is active or inactive), and the water overflow  $Q_i^{out}$  for each depression. These outputs enable the reconstruction of water reservoir distributions and the analysis of hydrological flows for comparison with observations. The hydrological model outputs are then post-processed to analyze the water volume distribution, water depth and accumulated outflow in each depression. To compute the water depth, the model uses the lake elevation  $Z_i$ , the ID label map and the elevation map. The ID label map is used to identify the area





covered by the depressions associated. The water depth map is then computed by subtracting the elevation of each grid cell of the depression i from the lake elevation. The water depth map provides insights into the water distribution across the planet and the topographic features of the depressions. The volume distribution map can be obtained by multiplying the water depth map by the area of each cell. The volume distribution map provides information on the amount of water stored in each depression. The accumulated outflow map is generated by mapping the overflow discharge  $Q_i^{out}$  for each depression, either normalized by the lake surface area or distributed across the contributing watershed area. While the model outputs the accumulated overflow  $Q_i^{out}$  for each depression, it does not explicitly resolve the flow pathways between depressions. To overcome this limitation and reconstruct a continuous river network, a post-processing procedure is applied. The first step consists in modifying the original topography by assigning the lake water levels  $Z_i$  to the corresponding depressions. This creates a new topographic surface where water can flow continuously across lake basins which overflow. Flat zones introduced by the filled lakes are then corrected using the 'resolve\_flats' function from the Pyshed library (Bartos, 2020), allowing proper flow routing. The flow direction is computed using the D8 algorithm (O'Callaghan and Mark, 1984), and flow accumulation is then calculated following the method of Freeman (1991). Based on this accumulation map, the main drainage paths are extracted with the 'extract river network' function of Pyshed. Finally, the overflow discharges from the hydrological model are assigned to the extracted river channels, producing a spatially explicit reconstruction of surface water pathways. This step provides a physically consistent representation of surface runoff and facilitates comparison with observed valley networks on Mars.

# 3 Application on Mars

#### 280 3.1 Mars topography

We used the topographic map from MOLA (Mars Orbiter Laser Altimeter Smith et al., 1999, 2001; Neumann et al., 2001) which covers all the Mars' surface with a fine resolution of 463 meters at the equator (1/128 degree per pixel, Figure 2a). This dataset provides a detailed digital elevation model (DEM) of Mars which represents approximately 1 billion cells (23,040 × 46,080 cells). This high-resolution topographic data is crucial for accurately identifying depressions and their watersheds. It is clear that the MOLA topographic map does not fully represent the topography of Mars during the periods when liquid water was stable on the surface. However, this current topographic map provides, on several preserved areas, mainly in the Southern Hemisphere, a good approximation of the past topography during the Noachian periods (4.1 to 3.7 billion years ago) when most of the fluvial features were formed (Tanaka et al., 2014). While the current MOLA topographic map accurately represents present-day Mars, it does not provide a global representation of the ancient surface topography required to study early Mars hydrology. Then, we also applied our model on a reconstructed topographic map of Mars built by Bouley et al. (2016) which remove the True Polar Wander (TPW) to better represent the global hydrology on early Mars. However, this reconstructed topographic map is only available at a lower resolution of 1 pixel per degree, which limits the ability to capture fine-scale topographic features and depressions.

**Figure 2.** Mars topographic map showing the inventory of observed data relating to water flow effects. (a) Digital Elevation Model (DEM) from the Mars Orbiter Laser Altimeter (MOLA) (Smith et al., 2001; Neumann et al., 2001). The shaded relief is generated from the DEM with a sun angle of 45° from horizontal and a sun azimuth of 315°, as measured clockwise from north. The blue lines represent the valley networks (Hynek et al., 2010). The white and red dots represent open-basin and closed-basin lakes, respectively (Goudge et al., 2016). The orange triangles represent deltas (Di Achille and Hynek, 2010). The magenta and yellow lines represent respectively Arabia (Perron et al., 2007) and Deuteronilus (Ivanov et al., 2017) shorelines. The white squares are zoomed areas on Nirgal Vallis' watershed (b), Jezero' watershed (c) and Gale' watershed (d).





Despite its limitations, the topographic analysis of MOLA data has provided significant insights into the hydrological activity on Mars, revealing a variety of geomorphological features indicative of past water flow (Figure 2a). Valley networks, such as those identified by Carr and Clow (1981) and Hynek et al. (2010), suggest the presence of fluvial processes that shaped the Martian surface, with drainage densities and morphologies pointing to sustained precipitation and surface runoff. Additionally, the identification of open- and closed-basin lakes, as documented by Goudge et al. (2016) and Fassett and Head (2008), highlights the role of topographic depressions in capturing and storing water. These lakes often exhibit inlet and outlet valleys, providing evidence of hydrological connectivity. Furthermore, deltas, such as those observed by Di Achille and Hynek (2010), mark the confluence of sediment-laden streams with standing bodies of water, offering clues about sediment transport and deposition processes. Finally, potential shorelines, including the Arabia and Deuteronilus levels described by Sholes et al. (2021), suggest the existence of ancient oceans in the northern lowlands.

Additionally, the recent crater database of 2020 (Robbins and Hynek, 2012) contains more than 385,000 craters with a diameter larger than 1 km. The density of craters is not uniform across the planet, with a higher density in the Southern Hemisphere and a lower density in the Northern Hemisphere (Figure 2a). This gives insights into the minimum number of leaf depressions that can be contained in the hydrological database.

To highlight the application of our model, we first present results at the global scale, then focus on three well-studied systems: Nirgal Vallis, Jezero Crater, and Gale Crater (Figure 2b-d). Nirgal Vallis (Figure 2b) is one of the longest valley networks on Mars, extending over 700 km. Its morphology makes it an ideal case study for understanding fluvial processes and the potential for past water flow on Mars. High-resolution MOLA data enable detailed hydrological simulations within this catchment. Jezero Crater (Figure 2c) and Gale Crater (Figure 2d) are of particular interest because they are the landing sites of the Perseverance and Curiosity rovers, respectively. Both craters host well-preserved deltas and sedimentary deposits such as conglomerates and stratified layers, offering strong evidence that they once contained standing bodies of water. By focusing our model to these sites, we can evaluate its ability to reproduce the formation of paleolakes.

## 3.2 Parameters exploration

In this section, we define the range of parameters explored in this conceptual application, including the evaporation rate E and the GEL. The evaporation rate E can be estimated using the following bulk aerodynamic formula:

$$E = \rho_{air} C_d v_{wind} \left[ q_{sat}(T_{surf}) - q_{air} \right]$$
(8)

where  $\rho_{air}$  (kg·m<sup>-3</sup>) and  $v_{wind}$  (m·s<sup>-1</sup>) are respectively the volumetric mass of air and the wind velocity,  $C_d$  is the aerodynamic coefficient (unitless),  $q_{sat}(T_{surf})$  is the water vapor mass mixing ratio at saturation at the surface which depends on the surface temperature  $T_{surf}$ , and  $q_{air}$  is the mixing ratio in the atmospheric layer. Assuming  $C_d = 2.5 \times 10^{-3}$ ,  $\rho_{air} = 0.02 \, \mathrm{kg} \cdot \mathrm{m}^{-3}$ ,  $q_{sat} = 0.01 \, \mathrm{kg} \cdot \mathrm{kg}^{-1}$ , neglecting  $q_{air}$  and by varying the wind velocity  $v_{wind}$  between 1 and 10 m·s<sup>-1</sup> (Wordsworth et al., 2015; Turbet et al., 2017), the evaporation rate E ranges from approximately 0.01 to 1 m·yr<sup>-1</sup>. This conceptual study proposes to test three evaporation rates: 0.01, 0.1 and 1 m·yr<sup>-1</sup>. Additionally, to test the robustness of this model, a non-realistic evaporation rate of 10 m·yr<sup>-1</sup> is also tested. Such value is much higher than the maximum evaporation rate




observed on Earth, which is around 3 m $\cdot$  yr<sup>-1</sup> (Houston, 2006). This additional evaporation rate will allow to evaluate the model behaviour under extreme conditions and assess its ability to handle large volumes of water.

Moreover, several GEL values are investigated. Present-day Mars is estimated to have a GEL of approximately 34 m (Carr and Head, 2015), while the late Hesperian era is associated with a GEL of 64 m (Carr and Head, 2015). Other studies suggest higher values, such as 137 m (Villanueva et al., 2015), 550 m (Di Achille and Hynek, 2010), and even up to 1,500 m (Scheller et al., 2021) and 1,970 m (Jakosky and Hallis, 2024). To encompass this variability, four representative GEL values are selected for this conceptual study: 1 m, 10 m, 100 m, and 1,000 m.

To evaluate the sensitivity of the hydrological model to initial conditions, this GEL values are distributed in three distinct initial water distributions: a homogeneous distribution across the planet (Figure 4a), in the northern lowlands (75°N, 60°W, Figure 4b), and in the Hellas crater (40°N, 60°E, Figure2c). These configurations allow us to assess the model response to varying initial states. Considering the tested evaporation rates, GEL values and initial states, a total of 48 simulations were performed to explore the parameter space.

## 3.3 Hydrological database

By using the MOLA topography map (Figure 2), the hydrological database was built on computing cluster (32 cores 2 AMD EPYC 7302 16-Core, 16GB of memory per core) in less than 2 days of computation time (41 hours equivalent to 1,312 CPU hours). The database contains 5,967,453 leaf depressions, 5,967,452 meta-depressions, i.e. a total of 11,934,905 depressions. The map of *ID* watershed and the watershed area  $A_i^w$  were used to build the Figure 3a. It shows the extent of the watersheds of leaf depressions near to the Gale crater (9,777 leaf depressions on this figure). This figure illustrates the high resolution of the database, higher than crater scale, which provides detailed information on the depressions and their associated watersheds. By looking at the craters on this map (circles contour plot, black lines), several depressions can be identified into them, including Gale crater, which is localized by the white star. The presence of depressions in the craters is due to local pits that are present in the craters.

As mentioned in the section 2.2, the hydrological functions are pre-computed for each depression. The hydrological functions of a random depression (i = 4281753, white point in the map figure 3a) which is close to the Gale crater, are shown in Figure 3b. The pre-computed lake volume  $V_i^l$  and lake area  $A_i^l$  are plotted as functions of the lake elevation  $Z_i^l$ . As expected, the lake volume monotonically increases with the lake elevation, while the lake area can have a flat or increasing trend. The lake area function can not strictly increase because the lake area can remain constant with the lake elevation when a lake is bordered by vertical cliffs for instance. The hydrological functions provide insights into the behavior of the depressions and their associated lakes, enabling the model to accurately simulate the water redistribution process.

Figure 3. (a) 9,777 watersheds are represented on the zoomed area of the Gale crater watershed (Figure 2d). Gale crater is localized by the white star. The color bar represents the watershed area  $A_i^w$ . (b) Example of pre-computed hydrological functions for a random depression (i=4281753) localized near to the Gale watershed by the white point (a). Evolution of the lake volume  $V_i^l$  (black line) and the lake area  $A_i^l$  (blue curve) as functions of the lake elevation  $Z_i^l$ .

# 3.4 Simulations and validation




Simulations were conducted on a supercomputer (32 cores 2 AMD Genoa EPYC 9654, 4 GB of memory per core), achieving a computational speed of approximately 22,000 iterations per day of simulation. A total of 48 simulations were performed to test the model sensitivity to the initial state (Figure 4a-c), the amount of water present on the planet and the fixed evaporation rate (Figure 4). The simulations were run for a maximum of 100,000 iterations. As shown in equations 4 and 5, the precipitation rate is directly linked to the lake area. A constant precipitation rate (i.e. a constant P/E ratio) indicates that the lakes are stable and that the model has reached a steady state (Figure 4d-g).

The convergence curves (Figure 4d-g) indicate that higher evaporation rates lead to faster convergence of the model. Even though the convergence time depends on the GEL and the initial state, it primarily depends on the fixed evaporation rate E and the adaptive time step  $\Delta t$ . The results also demonstrate that the model is not sensitive here to the initial state, as all simulations converge to the same P/E ratio for a fixed GEL. The Figure 4d-g illustrates, as demonstrated in Appendix A, that the final water distribution on the planet depend only on GEL value. For a fixed GEL, the simulations confirm that the model is able of redistributing water across the planet and converging to a unique steady state. As expected, the converged P/E ratio depends only on the total lake surface area which is directly related to the GEL. It is also observed that the higher the GEL, the higher the final P/E ratio, as the water surface has a larger exchange area with the atmosphere.

Figure 4. The three maps represent the initial states tested for a GEL = 100 m. The water is either distributed homogeneously (a), in the northern lowlands (b), or in the Hellas crater (c). The blue areas show the location of water reservoirs. The plots show the evolution of the P/E ratio over time for different GEL: 1 m (d), 10 m (e), 100 m (f), and 1,000 m (g). Solid lines, dashed lines, and dotted-dashed lines represent simulations with homogeneous water distribution (a), distribution in the northern lowlands (b), and in the Hellas crater (c), respectively. Green, orange, blue, and black lines represent simulations with fixed evaporation rates of 10, 1, 0.1, and 0.01 m yr<sup>-1</sup>, respectively.

# 3.4.1 Water distribution analysis


Figure 4 demonstrates that a unique steady state is reached for each GEL, regardless of the initial state and the evaporation rate. In this section, the global water distribution is analysed for the four GEL values. Figure 5 illustrates the distribution of water reservoirs (sky blue histogram) at steady state. The left column shows the distribution of water reservoirs per degree of latitude, while the right column shows the distribution per degree of longitude. The cumulative relative water volume (solid



black line) is compared to a conceptual homogeneous distribution of water (dashed black line) from the north to the south-pole for the left column and from the west to the east for the right column. The extent of main reservoirs in the northern lowlands (Arcadia, Acidalia, Utopia, Isidis and Vastitas Borealis) and the southern basins (Solis, Hellas and Argyre) are represented by the vertical colored dashed lines and are localized on the water depth map (Figure 6a).

**Figure 5.** Distribution of water reservoirs at steady state for each GEL: 1 m (a, b), 10 m (c, d), 100 m (e, f), 1000 m (g, h). The blue areas represent the proportion of water relative to the total water volume (left y-axis) per degree of latitude (left column) and longitude (right column). The solid black lines show the cumulative relative water volume (right y-axis). The dashed black lines represent the cumulative relative water volume for a conceptual homogeneous distribution of water.

For the simulation with the lowest GEL value (1 m, Figure 5a-b), the water is distributed relatively uniformly across the planet, with no significant regional accumulation (Appendix B1). According to latitude, the cumulative relative water volume (solid black line) closely follows the homogeneous distribution (dashed black line). However, there is a lack of stored volume between 40°N and 60°N, which is subsequently compensated by a larger storage between 0° and 15°S. This suggests that the region between 40°N and 60°N has fewer opportunities for retaining water, likely because it consists of relatively young, less-cratered terrains with limited storage capacity. As a result, water precipitating in this latitude band tends to flow toward




the northern lowlands. Additionally, because the model applies spatially homogeneous precipitation, the northern lowlands progressively lose water to supply downstream regions. In particular, the area between 0° and 15°S – characterized by older, heavily cratered terrain – acts as a more efficient water sink, capable of retaining substantial volumes. These two processes together explain the observed deficit at mid-northern latitudes and the accumulation at southern equatorial latitudes. In terms of longitude, the distribution appears relatively uniform, with a slight concentration between 60°E and 120°E. This concentration can be explained by water convergence from the north to the south due to the steep slopes imposed by the Tharsis dome. For this amount of water, the distribution is relatively homogeneous across the planet, showing little concentration or/and overflow of water. It can be concluded that this quantity allows for minimal overflow on a planetary scale.

The simulation with a GEL of 10 m (Figure 5c-d) emerges distinct peaks in the distribution of water. In terms of latitude, a concentration appears between 60°N and 90°N, corresponding to the Northern Lowlands. In comparison to the GEL of 1 m, the distribution of water is mainly changed in the Northern Hemisphere, where the water is concentrated in the Northern lowlands (Appendix B2). This concentration is due to an overflow of water from the areas between 60°N and 0° to the Northern lowlands. The cumulative relative volume curve is significantly different from the homogeneous distribution in the Northern Hemisphere. Approximately 20% of the total water volume is stored in the Northern Lowlands, while the Southern Hemisphere remains close to the homogeneous distribution, like the case with a GEL of 1 m. The distribution in the Southern Hemisphere is relatively uniform, with a slight increase between 0° and 15°S. This indicates that the model is able to store this amount of water in the Southern Hemisphere and the area between 0° and 60°N is not favorable for water storage. Longitudinally, the distribution is also pronounced, with peaks at 0° and 120°W, corresponding to Acidalia Planitia, and at 60° and 120°W, corresponding to the Hellas Basin. The cumulative relative volume curve is significantly different from the homogeneous distribution, mainly because of the water accumulation in the north of Acidalia Planitia and in the Hellas basin.

For a GEL of 100 m (Figure 5e-f), the distribution changes significantly. The Northern Ocean expands considerably between 30°N and 90°N, accounting for approximately 50% of the total water volume. Arcadia Planitia, Acidalia Planitia and Utopia Planitia are the main flowed areas which contribute to the extension of the Northern Ocean. A significant amount of water is also stored in the Hellas and Argyre Basins, accounting for approximately 20% of the total water volume on the planet, which corresponds in quantity to the water stored between 30°N and 30°S. The distribution according to longitude shows that the water in the Northern lowlands is mainly concentrated in Acidalia Planitia and Utopia Planitia, while Hellas, Argyre and Solis Basins are the main flowed areas in the Southern Hemisphere. Figure 6 illustrates spatially the main water reservoirs at steady state. As mentioned previously, the water depth map shows that the water is mainly stored in the northern lowlands and crater impacts, but it can also be stored in low slope areas with a natural barrier allowing to block the water flow. The zoomed maps show that all craters are filled with water due to homogeneous precipitation. Moreover, in addition to the craters, there are many local areas with thin water depth. In the zoomed maps, the valley network can also be distinguished, mainly the Nirgal Vallis (Figure 6b) and the valley network that feeds the Gale crater (Figure 6d).


**Figure 6.** Distribution of water reservoirs (blue areas) at steady state with a GEL equal to 100 m. The color bar represents the water depth. The brown shaded areas represent the areas where the water cannot be accumulated. The white squares are zoomed areas on Nirgal Vallis' watershed (b), Jezero' watershed (c) and Gale' watershed (d). The white stars localized the Nirgal Vallis outlet (b), Jezero crater (c) and Gale crater (d).

The simulation with the highest GEL value (1,000 m, Figure 5g-h) shows a significant increase in the volume of water stored in the Northern Ocean, which accounts for approximately 75% of the total water volume (Appendix B3). The Hellas Basin remains a significant reservoir, accounting for approximately 20% of the total water volume. The distribution according to latitude shows that the water is mainly concentrated in the Northern Hemisphere and in the Hellas Basin. The distribution according to longitude shows that the water is mainly concentrated in Acidalia Planitia, Utopia Planitia and Arcadia Planitia. The cumulative relative volume curve is significantly different from the homogeneous distribution, mainly because of the water accumulation in the Northern lowlands and in Hellas basin.



This analysis shows that between a GEL of 1 m and 10 m, a formation of a northern ocean begins to emerge. Overall, regions with limited water retention are found in the sparsely cratered zone between 30°N and 60°N and in the Tharsis dome area, where steep slopes promote rapid overflow and hinder accumulation. As the GEL value increases, the proportion of water stored in the Northern Hemisphere significantly rises. Similarly, an evolution in the water distribution is observed from west to east with increasing GEL values. Higher GEL values lead to a greater proportion of water being concentrated in the eastern part of the planet (0°E to 180°E).

#### 3.4.2 Water outflow analysis

At steady state, although water distribution across the planet depends on the GEL, the water outflow rate is primarily controlled by the precipitation rate P, which depends on the evaporation rate E and the GEL (i.e. oceans/lakes area  $A_i$ ), and whether the depression is filled to allow the overflow. The comparison between Figure 7 and the figures in Appendix B (Figures B2, B4 and B6) shows that the water outflow rate increases with the GEL, as the lake area increases with the GEL.

The water outflow map (Figure 7a) shows the water flow accumulation at steady state for a GEL of  $100 \,\mathrm{m}$  and an evaporation rate of  $1 \,\mathrm{m}\,\mathrm{yr}^{-1}$ . The water overflow  $Q_i^{out}$  is displayed on the watershed area for each depression where the lake overflows. This figure allows to keep a continuity between depressions and easily visualize the water flow pathways. In case there is no overflow, the lake areas are displayed by a transparent sky blue area. The white shaded areas represent the watershed area where there is no overflow. This result higlights four main valley networks that drain the water from the highlands to the Northern Ocean. Three of them finish their path in the part of the ocean corresponding to Acidalia Planitia, while the fourth one drains the water to the ocean corresponding to Arcadia Planitia. These valley networks are mainly located near the Tharsis dome, which is a highland area with steep slopes and no water reservoir to retain the water, that facilitates the water flow.

The highest simulated water outflow is observed near Marte Vallis (Amazonis Planitia). The outlet coordinates are located at 174°W, 42°N, with a discharge of approximately 66,034 m³ s<sup>-1</sup>. To the east of the outlet, the river drains Elysium Planitia, particularly the region south of Elysium Mons. To the west of the outlet, the river collects water from the entire Olympus Mons area and the western and southern regions of the Tharsis Montes. The second river with the highest flow is located at the junction between Simud Vallis and Lobo Vallis (coordinates 36°W, 35°N). This river drains the eastern part of the Tharsis dome and the Valles Marineris region. The flow rate is approximately 29,411 m³ s<sup>-1</sup>. The outlet of the third productive river is located near the previous one, at coordinates 30°W, 30°N, corresponding to Ares Vallis. This river drains the elevated and cratered region of Noachis Terra (between 30°W and 30°E) and the area situated between Valles Marineris and Argyre Planitia. The flow rate is approximately 12,362 m³ s<sup>-1</sup>. The last major productive river highlighted in this simulation drains the northern region of Tharsis. Originating in the Tharsis region, specifically near Ascraeus Mons, it flows northward, terminating at coordinates 65°W, 57°N. The flow rate of this river is approximately 9,176 m³ s<sup>-1</sup>. The discharge rates in this simulation are comparable to those observed in some of Earth's largest rivers. For instance, the Amazon River, the largest river on Earth by discharge, has an average flow rate of approximately 209,000 m³ s<sup>-1</sup> (Dai and Trenberth, 2002), which is significantly higher than the simulated rivers. The Congo River, with an average discharge of around 41,000 m³ s<sup>-1</sup>, is closer to the simulated outflows, particularly the Marte Vallis outlet. Similarly, the Ganges-Brahmaputra River system, with an average discharge of






approximately 35,000 m<sup>3</sup> s<sup>-1</sup>, and the Orinoco River, with about 32,000 m<sup>3</sup> s<sup>-1</sup>, also fall within the range of the simulated river discharges. Additionally, the Saint Lawrence River, with an average discharge of approximately 10,000 m<sup>3</sup> s<sup>-1</sup>, provides an example of a terrestrial river with a flow rate comparable to the last mentioned simulated river. These comparisons highlight that the modeled hydrological system could produce flow rates similar to some of Earth's major river systems.

Nirgal Vallis (Figure 7b) is part of the upstream section of the river system that ultimately flows into Ares Vallis, with a simulated discharge of  $3,700 \text{ m}^3 \text{ s}^{-1}$  at his outlet (white star). However, when compared to the observed river network map (Hynek et al., 2010) shown in Figure 2b, the simulated and observed river pathways do not spatially align. The simulated river drains a significantly larger area than what is observed, suggesting that the actual discharge of Nirgal Vallis may be lower than the modeled value in this simulation.

Jezero Crater (Figure 7c) is fed by two rivers converging from the north and northeast. The discharge of the northern river is 50.6 m<sup>3</sup> s<sup>-1</sup>, while the northeastern river has a discharge of 45.8 m<sup>3</sup> s<sup>-1</sup>. The volumes contributed by the two rivers are relatively similar. Closed lakes that do not overflow are also observed upstream of the Jezero system. These closed systems could occasionally act as reservoirs contributing to the inflow of Jezero Crater.

In the case of Gale Crater (Figure 7d), two river systems also converge, but with significantly different discharges:  $79.5 \, \text{m}^3 \, \text{s}^{-1}$  for the northern river and  $120.2 \, \text{m}^3 \, \text{s}^{-1}$  for the southern river. Unlike Jezero Crater, the confluence occurs well before the water reaches Gale Crater. Additionally, other smaller streams contribute between the confluence and the crater. The total discharge of the river entering Gale Crater is  $244.1 \, \text{m}^3 \, \text{s}^{-1}$ , highlighting the significant contribution of upstream flows to the crater hydrological system.

The proposed representation allows for the study of water exchanges between the watersheds of depressions. We propose to go further in the spatial representation of the results by adding a post-processing step to the hydrological model outputs to identify rivers and the exact pathways of water flow between lakes. The post-processing results presented in Section 2.4 are shown in Figures 7e-g. This representation allows to identify the exact paths taken by water and to delineate watersheds. While lakes are well represented spatially, the widths of the rivers displayed in this figure are not to scale but rather serve as a mapping of their locations. This post-processing will facilitate the comparison with observed valley networks on Mars and provide a better understanding of water exchanges between reservoirs.

Figure 7. Water overflow at steady state with a GEL = 100 m. The color bar represents the water flow accumulation. The areas where there is no outflow are symbolized by white shaded areas and sky blue areas represent the watershed and lake/ocean areas. The first line of zoomed plots shows the raw output data of the hydrological model which represents the water overflow associated to the watershed area. The second line of zoomed plots shows the water overflow associated to the valley network, after post-processing explained in section 2.4.







# 4 Discussions and perspectives

## 4.1 Topography dependence

The main limitation is the model dependence on the topography map which is a crucial input for the construction of the hydrological database, as it defines the watershed of depressions and their spillover point. The resolution of the topography map can significantly impact the model ability to accurately represent the hydrological features of the planet. We tested the model with the MOLA topography map degraded to 1 px/degree. The model was run with a GEL of 100 m and an evaporation rate of 1 m yr<sup>-1</sup>. The model is compared with results using the high resolution topography map (128 px/degree). The Figures 8b-c show the water distribution according to longitude and latitude. As Figure 5, the histogram shows the relative water volume per degree and the solid line shows the cumulative relative water volume. While the distributions of water reservoirs are relatively similar according to longitude, the distributions according to latitude are significantly different, notably in the Northern Hemisphere. The high resolution model transfers more water to the Northern lowlands, while the low resolution model shows a more uniform distribution of water across the area between 90°N and 0°. This difference can be explained by the fact that the smoothing of the topography map can increase the elevation of the spillover point and consequently increase the storage capacity of the depressions.

The same work was done with a pre-True Polar Wander (TPW) topography map (Bouley et al., 2016) (Figure 8a). This topography map contains significantly less depressions (craters) than the current MOLA topographic map and the Tharsis region is not represented. As expected, the pre-TPW model shows a significant difference in the distribution of water reservoirs compared to the model base on MOLA topography. If the pre-TPW topography accumulates less water in the Northern Hemisphere, around 45% of the total water volume is stored in lower latitudes, between -15° and 60°N, placing one of the main reservoirs closest to the equator than in the current topography (Figure 8c). In the Southern Hemisphere, the water is mainly concentrated between 0° and 45° in the Hellas and Argyre basins. According to the longitude (Figure 8b), we can observe an absence of water accumulation in the area between 30W° and 30E°, characterized by the flat curve of the cumulative relative water volume. In the Southern Hemisphere, this area corresponds to the Noachian highlands, which are preserved in the MOLA topography. The previous simulations with the MOLA topography show that these areas are favorable for water storage, as they are heavily cratered and contain many depressions. The absence of theses depressions in the pre-TPW topography, due to his resolution, leads to a significant decrease in the water storage capacity of this area. This result suggests that using the current topography map brings some challenges to understand early Mars. However, models of water surface flow on early Mars are limited by our ability to reconstruct the ancient surface topography.

To fully leverage the potential of such a hydrological model, it would be essential to use a high-resolution topography map that accurately represents the surface of early Mars. This would involve reconstructing the topography by removing the effects of the True Polar Wander (TPW) event (Bouley et al., 2016; Phillips et al., 2001). Additionally, the exclusion of younger craters that formed after the Noachian period would be crucial, as their presence introduces additional reservoirs that were not part of the ancient Martian landscape (Morgan, 2024; Liu et al., 2024). This reconstructed map would provide a more accurate representation of the ancient Martian surface, enabling better simulations of early hydrological processes and


facilitating comparisons with geological and geomorphological observations. An additional improvement for reconstructed ancient topography maps would be to artificially "recraterize" younger terrains. This could be achieved by statistically adding impact craters using a realistic size-frequency distribution, consistent with the expected crater population for the Noachian or Hesperian epochs. Such an approach would help restore the hydrological storage capacity and drainage patterns that would have existed prior to resurfacing events, thereby improving the fidelity of hydrological simulations on early Mars.

**Figure 8.** (a) Distribution of water reservoirs (blue areas) on pre-TPW topography (Bouley et al., 2016) at steady state with a GEL equal to 100 m. The color bar represents the water depth. The brown areas represent the areas where the water cannot be accumulated. (b-c) The solid black lines show the cumulative relative water volume per degree of longitude (b) and latitude (c). Three models are compared: the current MOLA topography (128 px/degree, orange line), the degraded MOLA topography (1 px/degree, blue line) and the pre-TPW topography (1 px/degree, black line).








# 4.2 Groundwater processes

In these results, we assume no infiltration ( $\beta=1$ ) and neglect subsurface flows. However, incorporating infiltration (Shadab et al., 2025) and groundwater flow processes could significantly improve the model ability to simulate the Martian hydrological cycle, offering valuable insights into the interactions between surface water and the Martian subsurface. This implementation would allow the simulation of processes such as aquifer recharge, subsurface flow pathways and groundwater discharge/sapping. Developing a global groundwater flow model, inspired by terrestrial hydrological frameworks (Fan and Miguez-Macho, 2011; Verkaik et al., 2024) and previous Martian studies (Luo and Howard, 2008; Horvath and Andrews-Hanna, 2021, 2017; Harrison and Grimm, 2005; Andrews-Hanna et al., 2010; Hiatt et al., 2024), could further enhance the integration of surface and subsurface hydrological processes to compare, for instance, the model ouputs with the U- and V-shaped valley networks profiles (Williams and Phillips, 2001) which result from the groundwater sapping and surface runoff, respectively.

#### 4.3 Climate coupling

In this conceptual study, the precipitation rate P, which depends on the evaporation rate E and the lake surface area, is considered homogeneous. This simplification allows the model to simulate a steady state, where the hydrological system reaches equilibrium. The steady-state results can be analyzed using metrics such as the X ratio (Matsubara et al., 2013, 2011) or the Aridity Index (AI) (Stucky de Quay et al., 2020). The X ratio, defined as the ratio of evaporated water to precipitated water, provides a quantitative measure of the hydrological balance in a region. Similarly, the Aridity Index (AI), calculated as the ratio of lake area to the total watershed area, offers insights into the spatial distribution of water and the relative humidity of different regions. These metrics also enable comparisons with previous works.

By coupling the hydrological model with a GCM, it becomes possible to explore the interactions between atmospheric circulation, spatial and temporal precipitation/evaporation variations. This coupling would provide a more comprehensive understanding of Mars' water cycle, enabling the study of dynamic processes such as the formation and disappearance of waterbodies according to the seasons. Additionally, the model could be extended to study transient hydrological states, such as the formation of large valleys in the Martian highlands caused by catastrophic paleolake overflows (Irwin III et al., 2004). These events, driven by sudden breaches in crater rims or other topographic barriers, could have significantly reshaped the Martian surface. By simulating such transient events, the model could help identify the conditions under which these features formed, linking them to specific climatic and hydrological scenarios.

The next goal of our work is to integrate this hydrological model into the Planetary Evolution Model (PEM) (Forget et al., 2024; Clément et al., 2024), a new modelling framework based on an asynchronous coupling with a GCM. The PEM can simulate long-term climate dynamics which allow us to study the impact of orbital-scale climate variations on the water cycle. In particular, we could investigate long-term accumulation/depletion of liquid water in lakes, including their cycle due to temperature and pressure changes, runoff after precipitation, the transport of water by rivers, the diffusion of water into the deep subsurface, etc. Moreover, the PEM can be used to set initial states for other simulations by determining steady-state realistic distribution of water for given orbital parameters.







# 4.4 Comparison tool

This model allows the simulation of oceans, seas, lakes and rivers on Mars, providing a global and detailed representation of the Martian surface flow. By coupling this model with a GCM, we will be able to compare simulation results with geological and geomorphological observations. For instance, we can compare the simulations with existing databases on open- and closed-basin lakes (Goudge et al., 2016; Fassett and Head, 2008), identified deltas (Di Achille and Hynek, 2010), potential shorelines (Sholes et al., 2021) and valley networks (Hynek et al., 2010). Additionally, the results can be compared with detected hydrated minerals (Bibring et al., 2006) and stratified sedimentary deposits (Malin and Edgett, 2000; Horvath and Andrews-Hanna, 2017), using lake levels and river discharges to estimate sedimentary deposits. Finally, the simulated river discharges can be compared with estimates of Martian fluvial discharges (Mangold et al., 2013), thereby validating and refining hypotheses on Mars' hydrological and climate evolution.

#### 5 Conclusions

We developed a global high-resolution surface hydrological model that explicitly exploits a pre-computed hierarchical depression graph and lake volume—area—elevation functions to simulate the redistribution of liquid water over planetary topography. Applied here to present-day and reconstructed Martian topographies, the model efficiently identifies storage areas (lakes, seas, putative oceans), their spillovers, and the integrated drainage pathways. The hydrological database (nearly 12 million depressions) and lookup strategy enable rapid iterative equilibration under simple forcing, providing a scalable framework for future climate coupling. Systematic exploration of Global Equivalent Layer (GEL) and evaporation configurations shows: (i) convergence to a unique steady-state water distribution that depends only on total water volume (GEL) and topographic structure under homogeneous forcing; (ii) the emergence of a contiguous northern ocean between GEL values of order 1-10 m; (iii) a progressive concentration of water storage in the northern lowlands, Hellas, and Argyre with increasing GEL (up to 75 % of total volume in the northern basin at 1000 m GEL); (iv) organization of four main trunk drainage systems funneling highland runoff toward Acidalia, Arcadia and adjacent sectors, broadly consistent with large-scale slope controls. The model further reproduces local-scale reservoir connectivity for sites of interest (e.g. Jezero, Gale), offering quantitative estimates of inflow partitioning and discharge magnitudes. While the model can reproduce many observed features, limitations remain, particularly regarding the current topography used and the absence of subsurface flows. Future work will focus on integrating these processes and coupling the hydrological model with a Global Climate Model (GCM) to explore the interactions between Mars' hydrology and climate. Additionally, the model robustness will be further tested by incorporating alternative topographic reconstructions. Overall, this work establishes a physically consistent yet computationally efficient foundation for simulating the large-scale organization of surface liquid reservoirs at planetary scale. It provides a bridge between geomorphological evidence and climate scenario testing, and a platform upon which progressively richer hydrological and climatic processes can be integrated to refine constraints on the planet's aqueous and atmospheric evolution.



Code and data availability. MOLA topography map can be retrieved from the NASA Planetary Data System PDS (Smith et al., 1999). The current version of the model is available under the CeCILL licence. The exact version of the model used to produce the results used in this paper is archived on repository under https://doi.org/10.5281/zenodo.17208793 (Gauvain, 2025), as are input data and scripts to run the model and produce the plots for all the simulations presented in this paper.

#### Appendix A: Analytical demonstration

The presented model is designed to reach a steady state where the water volume, lake surface area, and lake elevation in each active depression remain constant. To demonstrate the model ability to reach this equilibrium, an analytical demonstration is proposed to show that the filling of the different lakes depends only on the total available water volume and not on the assumed precipitation or evaporation rates. We consider a global hydrological network without subsurface connections, filled with a total water volume  $V_{tot}$  (m³) and subjected to an homogeneous precipitation P (m·s<sup>-1</sup>) and a potential evaporation E (m·s<sup>-1</sup>). There are N lakes, potentially connected (open lakes), within N watersheds of area  $A_i^w$  (m²). For each lake, the volume  $V_i$  (m³) is related to the lake area  $A_i$  (m²) by the function  $V_i = f_i(A_i)$ . The maximum volume  $V_i^{max}$  and the maximum lake area  $A_i^{max}$  at overflow are known for open lakes. At equilibrium, for an isolated closed lake i, the evaporation from the lake compensates for the precipitation falling on its watershed area:

$$EA_i = PA_i^w \quad \Rightarrow \quad A_i = (P/E)A_i^w \tag{A1}$$

For a system of N open lakes connected by overflow pathways, there is always a closed lake n located downstream of the system. Then, for all n, the Equation A1 can be written as:

$$E\left(\sum_{i=1}^{N-1} A_i^{max} + A_n\right) = P\sum_{i=1}^{N} A_i^w \quad \Rightarrow \quad A_n = \left(\frac{P}{E}\right) \sum_{i=1}^{N} A_i^w - \sum_{i=1}^{N-1} A_i^{max} = g_n\left(P/E\right) \tag{A2}$$

where g is a linear function, since both  $\sum_{i=1}^{N} A_i^w$  and  $\sum_{i=1}^{N-1} A_i^{max}$  are constant. Thus,  $A_n$  depends only on P/E, ensuring a unique relationship. This analytical result demonstrates that, at equilibrium, the distribution of water among the lakes depends exclusively on P/E. It can be noted that numerical artifacts depending on discretization (e.g., "vertical" lake shores) can make non-strictly monotonic functions. According to the equation A2, the total water volume stored in the lakes is given by:

$$V_{tot} = \sum_{i=1}^{N} V_i = \sum_{i=1}^{N} f_i(A_i) = \sum_{i=1}^{N} f_i(g_i(P/E))$$
(A3)

Since the functions  $g_i$  and  $f_i$  are strictly monotonic,  $V_{tot}$  depends only on P/E. It follows that for a given E and  $V_{tot}$ , there exists a unique corresponding value of P. This demonstration highlights the fundamental role of the GEL (i.e.  $V_{tot}$ ) in shaping the steady-state hydrological configuration of the system.

# Appendix B: Water distribution and water outflow

In this appendix, we present the water distribution and water outflow maps at steady state for different GEL values (1 m, 10 m and 1,000 m) with an evaporation rate of 1 m yr<sup>-1</sup>. These results complement those presented in sections 3.4.1 and 3.4.2 for a GEL of 100 m. The colorbars are consistent to facilitate comparison between the different results.

# B1 Global Equivalent Layer of 1 m

**Figure B1.** Distribution of water reservoirs (blue areas) at steady state with a GEL equal to 1 m. The color bar represents the water depth. The brown shaded areas represent the areas where the water cannot be accumulated. The white squares are zoomed areas on Nirgal Vallis' watershed (b), Jezero' watershed (c) and Gale' watershed (d). The white stars localized the Nirgal Vallis outlet (b), Jezero crater (c) and Gale crater (d).

Figure B2. Water overflow at steady state with a GEL = 1 m and  $E = 1 m.y^{-1}$ . The color bar represents the water flow accumulation. The areas where there is no outflow are symbolized by white shaded areas and sky blue areas represent the watershed and lake/ocean areas. The first line of zoomed plots shows the raw output data of the hydrological model which represents the water overflow associated to the watershed area. The second line of zoomed plots shows the water overflow associated to the valley network, after post-processing explained in section 2.4.

# B2 Global Equivalent Layer of 10 m

**Figure B3.** Distribution of water reservoirs (blue areas) at steady state with a GEL equal to 10 m. The color bar represents the water depth. The brown shaded areas represent the areas where the water cannot be accumulated. The white squares are zoomed areas on Nirgal Vallis' watershed (b), Jezero' watershed (c) and Gale' watershed (d). The white stars localized the Nirgal Vallis outlet (b), Jezero crater (c) and Gale crater (d).

Figure B4. Water overflow at steady state with a  $GEL = 10 \,\mathrm{m}$  and  $E = 1 \,m.y^{-1}$ . The color bar represents the water flow accumulation. The areas where there is no outflow are symbolized by white shaded areas and sky blue areas represent the watershed and lake/ocean areas. The first line of zoomed plots shows the raw output data of the hydrological model which represents the water overflow associated to the watershed area. The second line of zoomed plots shows the water overflow associated to the valley network, after post-processing explained in section 2.4.

# B3 Global Equivalent Layer of 1,000 m

**Figure B5.** Distribution of water reservoirs (blue areas) at steady state with a GEL equal to 1,000 m. The color bar represents the water depth. The brown shaded areas represent the areas where the water cannot be accumulated. The white squares are zoomed areas on Nirgal Vallis' watershed (b), Jezero' watershed (c) and Gale' watershed (d). The white stars localized the Nirgal Vallis outlet (b), Jezero crater (c) and Gale crater (d).

Figure B6. Water overflow at steady state with a  $GEL=1,000\,\mathrm{m}$  and  $E=1\,m.y^{-1}$ . The color bar represents the water flow accumulation. The areas where there is no outflow are symbolized by white shaded areas and sky blue areas represent the watershed and lake/ocean areas. The first line of zoomed plots shows the raw output data of the hydrological model which represents the water overflow associated to the watershed area. The second line of zoomed plots shows the water overflow associated to the valley network, after post-processing explained in section 2.4.


Author contributions. Alexandre Gauvain: Conceptualization, Methodology, Software, Validation, Visualization, Writing - original draft preparation, Writing - review & editing. François Forget: Conceptualization, Funding acquisition, Methodology, Project administration, Supervision, Writing - review & editing. Martin Turbet: Methodology, Writing - review & editing. Jean-Baptiste Clément: Methodology, Software, Writing - review & editing. Lucas Lange: Methodology, Visualization, Writing - review & editing. Romain Vandemeulebrouck: Methodology, Software.

Competing interests. The authors declare that they have no conflict of interest.

Acknowledgements. This project has received funding from the European Research Council (ERC) under the European Union's Horizon 2020 research and innovation program (Grant 835275) through the "Mars Through Time" project. To process the hydrological database, this study benefited from the IPSL Data and Computing Center ESPRI which is supported by CNRS, SU, CNES and Ecole Polytechnique. Simulations were done thanks to the High-Performance Computing resources of Centre Informatique National de l'Enseignement Supérieur (CINES) under the allocations n°A0140110391 and n°A0160110391 made by Grand Equipement National de Calcul Intensif (GENCI).

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
