# Peer review of "A Global High-Resolution Hydrological Model to Simulate the Dynamics of Surface Liquid Reservoirs: Application on Mars"

_EGUsphere, 2025_

## Referee Comment (RC1)

This paper presents a series of simulations of possible past water distribution on Mars, making use of a depression hierarchy and a hydrological model to distribute different possible amounts of water across the planet. It is great to see this work being done and I enjoyed reading the paper, especially the fantastic figures showing several theoretical water distributions for Mars. However, I do have several questions about the implementation of the hydrological model, pre-processing of the data, and conclusions around the final water distribution. Some of the questions that I had while reading the paper are listed as limitations of the model or discussed as directions for future work, but I felt as though more may be needed in the present paper. I hope that the authors will be able to address these questions and produce a stronger paper ready for publication.

**Major comments:**

**The hydrology model:** The authors use the Depression Hierarchy (DH) by Barnes et al to analyse the topography prior to routing water using a hydrology model. I have two major questions about their hydrology model:

- 1. Especially considering that DH was already used, why not continue and use Fill-Spill-Merge (FSM) by the same authors, which was constructed to route water through the DH? The authors refer to FSM and the methods used therein. Possibly there is some specific function that was not covered by FSM, but if so, it is not clear to me what this is. Especially given that the hydrology model was not explicitly coupled with any climate/GCM model, but was run iteratively with a calculation of E and P, the same thing could have been done with FSM with remarkably less effort on the part of the authors. It seems likely that I am just missing the reason here, in which case, it needs to be more clearly stated and in general, the differences between this and other similar algorithms/the specific need for this new algorithm needs to be stated (e.g. Noel et al 2021¹, Gailleton et al 2024², Shook et al 2021³).
  - 1. Line 213: the overflow algorithm described cites Barnes et al 2021 (FSM), and indeed seems to be identical to the FSM algorithm.
  - 2. Line 235: the bypass mechanism described herein already exists in FSM as well.
- 2. I am concerned about the pre-computed hydrological lake functions. I understand that the intention here was to increase computational efficiency, but I have not been convinced that it is worth it. Either a discussion of how much faster this is than FSM (or than the current model with a full calculation of lake levels), or actually calculating the real lake levels, would help here.

**The topography:** The authors mention in the text that it will be crucial for future work to exclude craters that formed after the Noachian period, ~3.7 billion years ago. I think that this issue is being rather undersold. The post-Noachian period includes most of Mars's history. Many, many craters could post-date this period. The inclusion of these craters in the current work is likely to create a significant bias in the results, particularly in the case of the low-GEL simulations, in which a significant portion of the water is likely stored in younger craters. I do not have enough knowledge of Martian geomorphology to speak to whether the larger craters also post-date this time period, but it is a concern. How meaningful are the results if the topography used is significantly different from the time period intended to be simulated?

**Validation:** I recognise that a comprehensive validation or evaluation of the actual distribution of water on Mars may be beyond the scope of this work, however, I felt that at minimum some comparison between the results and known locations of water on Mars was starkly missing. Figure 2 shows known data around lakes and deltas, and the paper computes stored water and overflow volumes. This seems like a perfect opportunity to, at minimum, compare the results between the different GEL values and

the known data and at least determine which simulation comes the closest to matching these. At a glance, the 100 m GEL simulation seems to come closest to the Deuteronilus and Arabia shorelines, and a deeper look into this would be helpful.

**Line comments:**

Line 65: I recommend citing DH at its first mention here. Similarly, this paragraph mentions the Mars topography, which should be cited here.

Line 114: the ocean elevation – the ocean elevation setting in DH does not fix the elevation of the oceans. Rather, it defines which cells should be considered as 'ocean' in order to start the process of locating pit cells to build the DH. The setting has no bearing on whether or not water redistributed through depressions can create a planetary scale ocean. I think, in your case, that choosing the highest cell on the topography should be okay in the workflow, but what is happening algorithmically then is that DH construction will initialise from the location of that highest cell and no more.

Line 119: I believe 'identify the depressions' should be 'identify the pit cells'.

Lines 180-185: My understanding is that a GCM was not actually used in this work, but some of the writing reads as though it was. Please be clearer that the use of a GCM is only a possibility for future work.

Equation 5: It is unclear to me why the total evaporated volume is being divided only by the areas of currently active watersheds. Shouldn't it be divided by the area of the planet, since precipitation is spread over the entire planet? In equation 6, similarly, the area of active depressions is used, and the text specifies active depressions. I know that you must be adding P into inactive depressions given that water distributions from all three starting points given in figure 4 converge. Or perhaps I am misunderstanding what an active watershed is (I thought that it was one that contained water in the last iteration).

Line 372: the unique steady-state reached for each GEL: This makes sense and I also think that it's simply an observation that when you use constant P-E rates over the whole planet, then water will distribute itself into the watersheds that have the largest contributing area. In other words, the multiple locations for setup of the starting water don't change the basic fact that, given enough time and, importantly, the globally constant P-E, depressions with the largest contributing area (and possibly smallest area-to-volume ratio) will pool all of the water. Given that DH provides both depression volumes and watershed volumes, I wonder whether you may have even been able to bypass the simulation entirely with some clever math (not sure if this would have been easier than just doing the simulation or not). (Note: I don't think you should remove the different starting distributions from the paper or anything, but possibly you could note that this result makes theoretical sense).

What I find more interesting is how different the distributions are with different amounts of total water (fig. 5), indicating some interesting relationships between depression level (leaf or higher) and contributing area. This seems like a great opportunity for some kind of validation, since the water distributions are so distinct with different GELs.

That said, given the effort to conserve mass in the computation of P based on E, here is a place where I am really concerned about the choice to not calculate lake volumes directly - mass will not be conserved with the linear estimate currently done.

**Figures:**

The figures are overall great, clear and informative. I just have a few comments to increase their clarity.

**Figure 1:** The dotted lines present on panel (b) are unclear. There are 4 dotted lines but only 3 numbers. I can't tell which line is supposed to correspond to which number; I know that the lowest line is the one that should not correspond to any number, but right now it looks as though it is intended to correspond to number 5.

**Figure 2:** Panels b, c, and d each contain a white star that I don't know the meaning of.

**Figure 3:** I assume that these are watersheds relating to the lowest, leaf-level depressions? Please clarify.

**Figure 8 and related text:** The text notes a lack of pooled water between -30 and 30 degrees longitude in the pre-TPW topography, and calls out its coarser resolution as the cause of this. However, the degraded MOLA topography still has water pooling in this region. This is not the only mismatch and generally, the full-resolution and degraded versions of MOLA match very closely (with one major exception noted in the text) while the pre-TPW topography looks significantly different. To me, this implies that spatial resolution is not the main causative factor, but rather that actual material differences in the topography are behind this. Comparing this figure with the distribution shown in Fig. 6, the main difference I can see visually is in the northern latitudes, where the MOLA topography has a vast ocean while the pre-TPW has virtually no water pooling. This also seems to be the cause of much of the difference in that -30 to 30 longitude zone. I can't see how such an expansive depression could be lost as a result of coarser topography — and indeed we know that the degraded MOLA still had water distributed in this region — so it would be great to see a discussion of that.

**Figure B2, B4, and B6:** These figures refer to a second line of zoomed plots which does not exist.

1S. A. Noel, A. C. Ault, D. R. Buckmaster, and J. V. Krogmeier, "A Rainfall-Based, Sequential Depression-Filling Algorithm and Assessments on a Watershed in Northeastern Indiana, USA," J. Adv. Model. Earth Syst., vol. 13, no. 6, 2021, doi: 10.1029/2020MS002362.

2B. Gailleton, P. Steer, P. Davy, W. Schwanghart, and T. Bernard, "GraphFlood 1.0: an efficient algorithm to approximate 2D hydrodynamics for landscape evolution models," Earth Surf. Dyn., vol. 12, no. 6, pp. 1295–1313, 2024, doi: 10.5194/esurf-12-1295-2024.

3K. Shook, R. Spiteri, J. Pomeroy, T. Liu, and O. Sharomi, "WDPM: the Wetland DEM Ponding Model," J. Open Source Softw., vol. 6, no. 64, p. 2276, 2021, doi: 10.21105/joss.02276.

---

## Community Comment (CC3)

[supplement omitted: unrelated document]